# Applying community health systems lenses to identify determinants of access to surgery among mobile & migrant populations with hydrocele in Zambia: A mixed methods assessment

**Patricia Maritim** [1]*, **Mwimba Chewe**[1], **Margarate Nzala Munakaampe** [1], **Adam Silumbwe**[1], **George Sichone**[2], **Joseph Mumba Zulu**[1]

**1** Department of Health Policy and Management, School of Public Health, University of Zambia, Lusaka, Zambia, **2** Participatory Research and Innovations Management, Lusaka, Zambia

☯ These authors contributed equally to this work.

* triciamarie20@gmail.com

**Data Availability Statement:** The datasets used and analyzed during the current study are available

## Abstract

Hydrocele which is caused by long term lymphatic filariasis infection can be treated through the provision of surgery. Access to surgeries remains low particularly for hard to reach populations. This study applied community health system lenses to identify determinants to the adoption, implementation and integration of hydrocele surgeries among migrants &mobile populations in Luangwa District, Zambia. A concurrent mixed methods design consisting of cross-sectional survey with hydrocele patients (n = 438) and in-depth interviews with different community actors (n = 38) was conducted in October 2021. Data analysis was based on the relational and programmatic lenses of Community Health Systems. Under the *Programmatic lens*, insufficient resources resulted in most health facilities being incapable of providing the minimum package of care for lymphatic filariasis. The absence of cross border collaborative structures limits the continuity of care for patients moving across the three countries. Other programmatic barriers include language barriers, inappropriate appointment systems, direct and indirect costs. In the *relational lens*, despite the key role that community leaders play their engagement in service delivery was low. Community actors including patients were rarely included in planning, implementation or evaluation of hydrocele services. Some patients utilized their *power within to* act as champions for the surgery but local groups such as fishing associations remained underutilized. Community health systems provide a potential avenue through which access amongst mobile and migrant populations can be enhanced through strategies such engagement of patient groups, knowledge sharing across borders and use of community monitoring initiatives.

within the manuscript and as supplementary materials.

**Funding:** This publication was supported by a grant to JMZ from the United States Agency for International Development (USAID) and UK FCDO from the British people (UK FCDO) through the Coalition for Operational Research on Neglected Tropical Diseases (COR-NTD) and administered by the African Research Network for Neglected Tropical Diseases (ARNTD). The funders had no role in the study design, data collection and analysis, decision to publish or preparation of the manuscript. Its contents are solely the responsibility of the authors and do not necessarily represent the views of USAID, UK FCDO, COR-NTD or the ARNTD.

**Competing interests:** The authors have declared that no competing interests exist.

## Introduction

Lymphatic filariasis (LF) is a Neglected Tropical Disease (NTD) caused by parasitic worms *Wuchereria bancrofti*, *Brugia malayi* and *Brugia timori*. An estimated 893 million people are at risk of infection globally [1]. If left untreated, long term infection usually leads to chronic manifestations such as lymphoedema, elephantiasis, acute inflammatory episodes termed dermato-lymphangioadenitis (ADLA) or hydrocele that require specialized care and treatment [2]. In highly endemic regions, genital manifestations, specifically hydrocele are the most common chronic forms of the disease. Hydrocele occurs when the tunica vaginalis sac around the testis increases in size due to fluid accumulation as a result of filarial infection [3]. Over 25 million men around the world exhibit symptoms of hydrocele [1,2,4]. The condition negatively affects the extent to which patients can be involved in activities within their homes, workplaces or communities [5].

The World Health Organization recommended Minimum package for Morbidity Management and Disability Prevention (MMDP) services for LF consists of mass drug administration to interrupt disease transmission, hydrocoelectomies (surgery), prevention of lymphoedema and ADLA [6]. Hydrocelectomies performed at district, higher level hospitals, or surgical camps [3,6] result in positive surgical outcomes including enhanced patient quality of life or reduction in scrotum size [6,7]. Despite the effectiveness of these surgeries their uptake remains low [8,9]. Implementation of hydrocelectomies and other NTD interventions has often relied on broad approaches aimed at universality which results in health inequities stemming from multiple pathways [10]. First, limited access to services causes differences in exposure and vulnerability to disease which results in poorer health outcomes [9]. Secondly, utilisation of services has varied social and economic consequences some of which are catastrophic for different categories of patients. Thirdly, delayed access leads to unequal treatment outcomes which has an impact on patients' social positions within their communities [11]. In some communities, hydrocele patients tend to have a high social standing as they are perceived to be well respected and wise. Consequently, patients may decide not to have surgery due to fear that their social standing will change [9]. Studies examining populations left behind during implementation of NTD interventions have identified different disadvantaged and hard to reach populations including; migrants/mobile populations, urban populations, communities in conflict areas, ethnic minorities, young girls and women of reproductive age, out of school children and children in private schools [12–14].

Mobile and migrant populations face unique health needs due to their high level of mobility which makes them more likely to be poorly connected to static public health and surveillance systems [15]. These challenges include; inability to afford the transport costs, limited awareness of existing services, sociocultural and gender norms that affect the ease with which communities can talk about their sexual health, privacy and confidentiality, fear of accessing services because they are not legally registered, feelings of discrimination, refusal of health services by service providers and service unavailability at nearby health facilities [16–18]. Additionally, inadequate representation of these populations in political processes and decision making means that their needs are often left out of health policy agenda including those aimed at improving access to healthcare [16,19]. This culminates in poor reach of health promotion and community outreach services aimed at case identification and referral into formal health care systems. Due to the transient nature of the populations, it is difficult ensure high quality of services because of logistical difficulties, poor infrastructure and limited training and capacity building for health care providers [16]. Migrant populations pay higher health service fees when they go to health facilities in the district. Additionally, language barriers also impede effective history taking that is a challenge for proper case identification particularly by

Community health workers identify and refer patients to nearby health facilities. Further, gaps in the integration of MMDP services into the community health system (CHS) has reduced their compatibility to community health systems and hampered community participation which poses a challenge to disease elimination [20–23].

## Theoretical framework

CHS are defined as "the set of local actors, relationships, and processes engaged in producing, advocating for, and supporting health in communities and households outside of, but existing in relationship to, formal health structures" [24]. Under CHS thinking, implementation processes often extend to social processes in the community that bring together different actors whose interactions can affect more upstream health outcomes such as awareness raising of available services, tackling social norms and meeting universal health coverage goals [16,25–27].

CHS has four main lenses; the programmatic, relational, collective action and critical lenses [27]. The *programmatic lens* views CHS as a part of the continuum between primary health care systems and the CHS itself within a bound geographical space with a set actors, populations and programmes based on existing policies and plans usually with determined goals and outcomes [27,28]. The *Relational lens* is interested in the actors, their formal or informal relationships and the flow of power within communities which are complex, dynamic and adaptive [27,28]. This lens explores the cognitive processes these actors go through to make sense of an intervention during its implementation and the ways in which they reinvent the intervention to fit within local contexts [27,29,30]. The *collective action* lens examines the mechanisms and processes that enable the different actors within the CHS to mobilize, collaborate and act collectively on health [27]. Finally, the *critical lens* places the CHS within the wider social, political and economic environment that forms the context for the health sector [27]. The critical lens seeks to map power relations through an assessment of ideas, interests and institutions an their interplay in the delivery of health services through approaches such as political economy analyses, decolonization or feminist and intersectional theories [27]. We have found that embedding MMDP interventions within CHS is effective in improving patient-centered access to LF care in Zambia [25]. Thus the main aim of our study was to use community health system lenses to identify CHS determinants which can be targeted using implementation strategies [31] to enhance the adoption, implementation and integration of hydrocele surgeries into local communities in an endemic district in Zambia with a focus on migrant and mobile populations.

## Materials and methods

### Study design

We conducted a concurrent mixed methods design [32] through a descriptive cross sectional survey with hydrocele patients and a qualitative case study with community actors including patients from mobile and migrant populations in October 2021.

### Study setting

Lymphatic filariasis is endemic in 75% of all districts in Zambia and more than 12.2 million people are at risk of infection [20]. Though morbidity mapping of hydrocele patients is done as part of national mass drug administration campaigns, the numbers are not an accurate reflection of existing cases as shown in our previous work [9]. The health seeking practices among patients in the districts has been described elsewhere [9,25]. Nonetheless, the health needs of mobile, fishing and migrant populations commonly found in Luangwa district and

Zambia in general have not been well defined in existing literature. The district is highly endemic for Lymphatic filariasis [33,34] and it shares a border with Mozambique and Zimbabwe. The district is predominantly rural with the main income generating activities in the area being fishing on the rivers, production of reed mats and subsistence farming. The local population is served by 14 Rural Health Centers and 3 second level (district) Hospitals. Hydrocele surgery is typically provided at the district hospital after referral from the lower level health centres.

A pilot initiative implemented by the University of Zambia and the Luangwa District Health Office between January 2019 and March 2020, supported the provision of the basic WHO MMDP package to all the health facilities in the district including hydrocele surgeries [25]. Findings from the summative evaluation of the initiative found improved patient access of up to 40.08% and a significant improvement in patient quality of life. Despite this increase, 25% of the hydrocele patients did not access any services during the intervention period, most of whom were from hard-to-reach populations including fishermen, farmers and migrants as the district borders Mozambique and Zimbabwe.

## Cross sectional survey with hydrocele patients

Our work in the district has found that community health workers (CHWs) are instrumental in case identification and referral of patients but morbidity mapping exercises are not often done due to inadequate funding for NTD interventions [25]. Consequently, to update the district record of patients, our data collection exercise doubled up as a morbidity mapping exercise (complete enumeration) extending to fishing camps along the river Luangwa as well as communities whose boundaries cross over into Mozambique and Zimbabwe. It is common for communities living in the district particularly those close to the borders to move freely across the three countries due to farming, trade or for their families. CHWs who had received MMDP training went household to household to identify new and old patients all of whom were invited to take part in the study. They identified a total of 438 hydrocele patients, all of whom took part in the survey. Questionnaires were administered by the CHWs who had received prior training. The questionnaires captured information on demographic and socio-economic characteristics, current disease management practices, the mental health impact of the disease on patients well as individual and community health system barriers that hamper their ability to access appropriate care.

## In-depth interviews

Qualitative data was collected through in-depth interviews with hydrocele patients including migrants and fishermen (n-20) and community actors such as CHWs, leaders, and healthcare providers (n = 18). Sampling was done through purposive and convenient sampling. Participants were invited to take part in the study either through a telephone call or face-to-face. Data was collected at locations convenient to the participants including within households or health facilities. Semi-structured interview guides were used to collect information on key CHS determinants and potential implementation strategies. The duration of each interview ranged from 45 mins to 1 hour 30 minutes. The interviews were done in English, Nyanja, Chewa and Bemba by multilingual research assistants who were also responsible for transcription and translation. All data collection tools used in the study are provided as S1 File.

## Data management and analysis

Quantitative data was collected using tablets and stored on an online server. Upon completion of data collection, each questionnaire was checked for errors and completeness. The completed

and checked dataset was then downloaded in Excel format and stored on a shared drive. Quantitative data analysis focused on descriptive statistics in the form of tables and graphs. The tables and graphs were generated using STATA15 software and Microsoft Excel. To understand the distribution of characteristics across respondents, the descriptive tables included the demographic characteristics of the study participants, their management practices of hydrocele and their perceptions on their ability to access health care as well as their perception of the health care system capacity to manage hydrocele.

Qualitative data was audio recorded, transcribed verbatim, entered QSR NVIVO version 12 for management and coding. We applied two lenses of Community Health Systems based on the definitions proposed by Schneider et al; the programmatic and relational [27] to guide analysis and interpretation. These frames were considered appropriate for our study of mobile and migrant populations whose mobility requires a definition of community that is not merely geographical but is also associated with shared identities and networks that extend beyond locality associated boundaries [27]. Further, as hydrocele services are currently not tailored for mobile and migrant populations, we felt that the mapping the system components and how they interact would form the foundation for future collective action. We used a combination of deductive approach, moving between understanding the connection between the nature of the problem (health inequities) and identifying potential solutions (CHS determinants and implementation strategies) that may lead to desired outcomes (equitable implementation and patient outcomes for mobile and migrant populations) [35,36]. Our qualitative and quantitative study results were integrated on the basis of an additional coverage approach [36]. Both quantitative and qualitative data had equal, substantial yet different contribution to answering the research questions leading to a more comprehensive answer to our main study aim. The reporting of findings was aligned to the GRAMMS checklist provided as S2 File.

### Ethical approval and consent to participate

Ethical approval was obtained from ERES Converge IRB (Ref. No. 2021-Jan-006) and the National Health Research Authority (Ref No: NHRA00012/4/03/2021). Study participants provided written consent prior to taking part in the study.

## Results

### Demographic characteristics of survey participants

A total of 438 hydrocele patients took part in the survey. More than half of the patients were aged 40 years and above (62.1%) and were married (65.6%). A quarter of the participants had either never attended school (25.9%) or the highest level of education they had attained was primary education (48.9%). In terms of employment, 76.6% of the participants either did not have formal employment work or worked within the home. Of those who worked, the main sources of income were either farming (62%) or fishing (26.6%). A majority of patients (76.09%) were from Zambia whereas 8% were from Zimbabwe and 16% were from Mozambique. Less than 1 percent (0.69%) of patients in the survey were from other countries. Though 79% of the patients have a permanent residence in Zambia, 54.4% reported moving across the borders seasonally for activities such as trading, fishing or farming as illustrated in Table 1.

### Clinical condition and morbidity management practices

Most of the patients had unilateral swelling (80.6%) and had begun exhibiting symptoms within the last ten years. A small proportion of the patients (15.5%) had swelling on both the left and right scrotum. There was a high level of awareness of existing health services for the

**Table 1. Demographic characteristics of survey participants.**

| Variables | Frequency | Percentage |
|---|---|---|
| | *(N\* = 438)* | *(%)* |
| **Age Group** | | |
| >19 | 40 | 9.13 |
| 20–24 years | 21 | 4.79 |
| 25–29 years | 33 | 7.53 |
| 30–34 years | 28 | 6.39 |
| 35–39 years | 43 | 9.82 |
| 40–49 years | 103 | 23.52 |
| 50–59 years | 82 | 18.72 |
| 60 + years | 88 | 20.09 |
| **Nationality** | | |
| Zambia | 331 | 76.09 |
| Zimbabwe | 33 | 7.59 |
| Mozambique | 68 | 15.63 |
| Other | 3 | 0.69 |
| **Do you have a permanent residence in Luangwa** | | |
| Yes | 329 | 79.09 |
| No | 87 | 20.91 |
| **Do you travel across the borders seasonally?** | | |
| Yes | 197 | 54.42 |
| No | 165 | 45.58 |
| **Marital Status** | | |
| Single | 124 | 28.31 |
| Married | 278 | 63.47 |
| Divorced/Separated | 21 | 4.79 |
| Widowed | 9 | 2.05 |
| Cohabiting | 2 | 0.46 |
| Have other wives | 4 | 0.91 |
| **Education Level** | | |
| No education | 113 | 25.92 |
| Primary School | 213 | 48.85 |
| Junior High School | 41 | 9.4 |
| Secondary School | 62 | 14.22 |
| Vocational training | 1 | 0.23 |
| University/College | 5 | 1.15 |
| Other (Specify) | 1 | 0.23 |
| **Disability** | | |
| Yes | 146 | 34.03 |
| No | 283 | 65.97 |
| **Main economic activity** | | |
| Employed | 25 | 5.9 |
| Looking for a job | 74 | 17.45 |
| Household work | 162 | 38.21 |
| Does not work | 163 | 38.44 |
| **Main income source** | | |
| Farming | 170 | 62.04 |

*(Continued)*

**Table 1.** (Continued)

| Variables | Frequency | Percentage |
|---|---|---|
| | *(N\* = 438)* | *(%)* |
| Fishing | 73 | 26.64 |
| Day worker | 6 | 2.19 |
| Small scale enterprise/self-employed | 9 | 3.28 |
| Private sector employment | 2 | 0.73 |
| Civil servant/ Government official | 1 | 0.36 |
| Other (Specify) | 13 | 4.74 |

*Missing values and not applicable responses excluded.

condition with information being provided mostly by community health workers (57.8%) or family members (21.4%). More than half of the patients (56.9%) had sought treatment at the health facility within the 12 months preceding the survey and 47.6% who had sought the services of a traditional healer (Table 2). Of the participants that had visited a health facility in the 12 months preceding the study, 63% indicated that they had received medication during their visit whereas only 2% reported having had surgery.

## Community health system determinants shaping access to hydrocelectomies

**Programmatic lens.** The programmatic lens views the CHS as "a bounded geographical space, consisting of a defined set of actors, populations and programmes, established through national policies and plans with measurable goals" [27]. The implementation of LF services including hydrocelectomies in is guided by the National NTD Masterplan which outlines service delivery strategies appropriate for facility and community settings [33]. Specific targets relevant to the provision of hydrocelectomies include full coverage of MMDP services in all implementing units and ensuring that 75% of all hydrocele patients having received surgery by 2023 either through health facilities or mobile outreach camps [33]. Despite this goal, 42.4% of survey participants had not sought medical assistance for their condition in the 12 months preceding the survey due to the reasons described below.

*Limited resources to provide hydrocelectomy services.* There was insufficient resources for the provision of hydrocele services due to limited district allocation which resulted in facilities failing to provide the minimum MMDP packages. This was characterized by challenges in procurement of surgical and medical supplies, creation of designated spaces for patients where they can feel comfortable and support for health promotion activities conducted at all levels within the catchment areas. For instance, lack of transportation hampered the ability of CHWs to effectively conduct case identification and patient follow-up post operation. Passive case identification strategies that were previously being used district had led to underestimation of the number of cases as most patients would go to traditional healers before registering at nearby facilities.

*Low levels of acceptability.* There was high level of awareness of the availability of surgeries in the district (89% of survey participants). However, the absence of cross border collaborative structures that can support health promotion activities, patient hand over or mobile outreach camps also limited the continuity of care for the patients moving across the three countries. A majority of the patients from mobile and migrant populations (79.4%) could not access services when travelling. The acceptability of the surgeries in the district was identified as a key

**Table 2. Disease status and morbidity management practices among hydrocele patients in Luangwa District.**

| Variable Name | Frequency | Percentage |
|---|---|---|
| | (N* = 412) | (%) |
| **How long have you had Hydrocele** | | |
| Less than 1 year | 8 | 1.94 |
| 1 to 3 years | 192 | 46.6 |
| 4 to 6 years | 94 | 22.82 |
| 7 to 10 years | 40 | 9.71 |
| 11 to 20 years | 46 | 11.17 |
| 21 to 30 years | 13 | 3.16 |
| 31 to 40 years | 13 | 3.16 |
| Over 40 years | 6 | 1.46 |
| **Does the swelling appear bigger on one side or is it evenly spread out** | | |
| One side (unilateral) | 349 | 80.6 |
| All sides are swollen to the same degree (bilateral) | 84 | 19.4 |
| **Do you have hydrocele on the left side, right sides or both sides?** | | |
| Left sides | 186 | 42.96 |
| Right sides | 180 | 41.57 |
| Both sides | 67 | 15.47 |
| **Are you aware of any health care services that can help treat your condition?** | | |
| Yes | 366 | 85.71 |
| No | 61 | 14.29 |
| **How did you find out about the existence of these services?** | | |
| Family members | 78 | 21.43 |
| Friends | 56 | 15.38 |
| Community meetings | 19 | 5.22 |
| Community health worker/Health care provider | 203 | 55.77 |
| Radio, TV or social media | 4 | 1.1 |
| Community leaders | 3 | 0.82 |
| Other (Specify) | 1 | 0.27 |
| **Have you ever sought medical help for your condition from a health facility in the past 12 months?** | | |
| Yes | 243 | 56.91 |
| No | 181 | 42.39 |
| **What type of services did you receive?** | | |
| Surgery | 4 | 1.68 |
| Draining of fluid | 28 | 11.76 |
| Medication | 149 | 62.61 |
| Referral to another facility | 10 | 4.20 |
| Did not receive any form of care | 17 | 7.14 |
| Other specify | 30 | 12.61 |
| **Have you ever sought medical help for your condition from a traditional healer in the past 12 months?** | | |
| Yes | 194 | 47.55 |
| No | 214 | 52.45 |

*Excludes participants below the age of consent (below 18), Missing values excluded.

challenge. According to the health providers, it was common for patients who had small hydroceles to delay accessing care until their condition has progressed significantly, and the pain became unbearable. Other reasons why patients shunned the services included myths about the disease being caused by witchcraft, preference for traditional healers, fear of the surgery or its consequences (sterility or death), fear of being attended to by female health providers and fear of health providers disclosing details of their condition to other community members. A community leader shared one of his experiences with a migrant patient;

> "I once took a hydrocele patient from Mozambique to the hospital so that he can access hydrocele services. He ran away from the hospital because he was afraid of dying after surgery. The second time he was taken to the district hospital he ran away again before accessing the services because he was still afraid of dying as result of surgery." (Community Leader, IDI 1).

*Fear of deportation among migrant populations.* In relation to the appropriateness of services, for migrants who come into the district for shorter periods of time and are issued with border passes, there was a worry that if they agreed to the surgeries their passes would expire while they were still in hospital or that if they had any post-operative complications, they would not have access to care while on the move. Health providers pointed out that it was common to see migrants who did not have all their legal documentation delay their treatment because they did not want to risk being reported to immigration officials by facility staff.

> "They do know that they are supposed to go to the hospital when they suspect hydrocele. But maybe the only thing that stops them is asking them questions and maybe staying for a long time when they are admitted at the hospital and their pass might expire, no relatives to wait for them. This is mainly for migrants." (Community Health Worker, IDI 6).

*Language barriers.* Language barriers especially for fishermen, farmers and migrants from Mozambique who spoke Portuguese affected the extent to which they could communicate with health providers and community health workers with ease.

> "It is easy to deal with some people coming from Zimbabwe maybe because they speak a bit of English and I speak English. But our friends from across (Mozambique), they cannot speak the local languages like Chikunda. It means when they come they speak Portuguese here which I have no knowledge of, so may be one barrier is language." (Community Health Worker, IDI 1).

*Rescheduling of appointment systems discouraged patients.* Appointment systems whereby patients were designated specific days and times when they could get the surgery though efficient was a hindrance if not adhered to. Patients, particularly the fishermen, felt discouraged from accessing services when they went to facilities and their appointments were rescheduled. Especially if it was during the fishing seasons and they had taken time off which translated to loss of income.

> "These patients may go there and they say, we have given you another day which you have to come back. Those people have things to do so this contributes to them missing the appointments. The day when the patient is ready, the people of the hospital will also be busy, and you will find that they may give them one day like all hydroceles patients should report on Wednesday. Then that patient on Wednesday won't be there." (Community Health Worker, IDI 5).

*Costs associated with the hydrocelectomies.* Direct and indirect costs associated with utilizing the surgery were a key impediment to 46% of the survey participants of whom 72.3% were not able to cover the costs at all. Residents did not pay for services at nearby health centres from where they received referrals to one of the three hospitals. Without referral letters, patients who went directly to the hospitals paid a fee. Patients from Zimbabwe and Mozambique paid fees that are higher than local patients. For fishermen and traders, the loss of income associated with the surgery and recovery durations, forced them to opt out of the surgery. Transportation costs were prohibitive for the migrants as the three hospitals with trained hydrocelectomy teams are not close to the common entry points into the district from Zimbabwe and Mozambique. Furthermore, costs related to food and having support systems such as family and friends around during the surgeries were also viewed as barriers for migrants.

> "The other reason is sometimes I do think of going back to the hospital for operation and it requires money to pay which I don't have. Then I may need to be admitted in hospital for some days as I may not be released immediately after the operation and then at home how will the children survive without me?" (Patient, IDI 12).

The regularity of monitoring and evaluation of the implementation of hydrocele services was linked to availability of resources. Community monitoring activities which provide an opportunity for joint evaluation of services and relationship building through reflexive learning, usually through neighborhood committee meetings or by using community scorecards, were rarely held.

**The relational lens.**   The relational lens considers the CHS as "a 'peopled' system of relationships–formal or informal–involving a wide variety of actors, interests and expressions of power that together constitute a social system" [27]. The expressions of power and their impact on how hydrocele services are delivered were categorized either as power with others, power to act and power within [37,38]. Previous and current implementation of hydrocelectomies in Luangwa district has typically involved; district health officials, health care providers at local facilities, community health workers, neighbourhood health committees, chiefs and village headmen, traditional healers, implementing partners such as the University of Zambia School of Public Health, patients and their families.

*Community leadership and influence on demand creation.* There were various instances where the *power to act* was evident in how the actors applied their understanding of the local context to support the delivery of services in the district. First, Community leaders used their respected positions to champion for the interventions and spoke publicly in different fora about the disease and the value of the surgery thereby challenging the secrecy surrounding discussions of hydrocele. By sharing their personal experiences with the surgeries, community leaders who were past hydrocele patients, promoted the service as an effective intervention in a district where hydrocele is considered a normal heritable disease and where the first line of treatment is to go to traditional healers. The leader's motivation stemmed from a concern that community development will be stalled if men especially young men continued to be excluded for social and economic activities that could benefit their communities. After case identification exercises, some of the leaders asked the CHWs to appraise them of the burden of hydrocele within their catchment areas to facilitate mobilization. Further, they were also able to hold meetings with fishermen coming back from fishing expeditions or encourage migrants to seek services within local health facilities without fear of being discriminated against or denied services. Nevertheless, the level of engagement of traditional leaders (14.6%) and religious leaders (17%) was quite low and their engagement presents a potential avenue to improve access.

"Fishermen usually come back to the village on Sundays and as the headman of the village I would and announce and talk to them about Covid 19 as well [as] teach them about hydrocele." (Community leader, IDI 1)

"That fear is there because some patients think that since they are foreigners no one would help them. But when they come to Zambia community leaders in the villages would tell the migrants to just go to the hospital and access available services there is no discrimination in accessing hydrocele services." (Community health worker, IDI 5)

*Community ownership of hydrocelectomy services.* Community health workers' familiarity with their catchment areas positioned them as intermediaries for the implementation of generalised services for hydrocele by sharing information received during the training at community meetings or within households, conducting patient identification, making referrals and conducting patient follow ups post operation. For mobile and migrant patients who lived in villages where community boundaries went into Zimbabwe and Mozambique, CHWs would cross over to identify patients and refer them to care. Participants described strategies that the CHWs would employ to ensure the reach of key hydrocele related health messages.

*Support from local Ministry of Health officials.* Thirdly, district health officials used their positions to influence the acceptability and adoption of the surgeries among the patients. For instance, their involvement in community mobilization and sensitisation exercises, increased the legitimacy of the health services and dispelled some of the fear and mistrust that patients had causing them to come in for verification and referral to surgery.

"There was a day that we were going around in the villages with the district staff distributing these drugs and showing pictures of the swelling scrotum, lower limbs, that was when he saw us and realized that for these to come and show us these pictures, there must be something that can be done." (Health provider, IDI 4)

*Collaboration among various community actors.* The extent to which the actors were able to exercise their *power with others* through forging relationships that would foster collaborative action aimed at improving access to hydrocele services was not very pronounced. Slightly, more than half of the patients (54.2%) in the survey felt that the different community actors collaborated effectively in delivering the hydrocele services. Additionally, the patients stated that health providers (74.8%), Community health workers (71.6%), Community based groups (33.79%) and Neighbourhood Health Committees (32.3%) played the greatest roles in implementation of services. Community members and patients are rarely included in planning, implementation or evaluation of hydrocele services. Community members expressed their social support to patients through contributions to the costs of accessing care where necessary. Moreover, other community actors drawn from local organizations, non-governmental organizations, government ministries or community groups were minimally engaged in the implementation of hydrocele and other MMDP services.

Only 38.7% of patients in the survey had ever been included in the different implementation phases. For patients who were utilizing their *power within to* act as champions for the surgery among other patients such exclusion was a missed opportunity to enhance the acceptability of the services. Further, the absence of patient groups in the district was an under-utilized mode through which patient engagement could be enhanced. For fishing populations, fishing associations who were currently not being engaged in service delivery, represented another opportunity for harnessing the *power within*. As the existing structures could be used to encourage patients to not seek care but they could also work with the other actors to

help set up surgical camps close to the fishing camps through which the surgeries could be brought closer to their members.

> "With the community, there is an area where the fishermen always start from before they go and fish and they have an association, so if health education can be given to them on what is, what causes and what are the benefits of hydrocele can be given to them, it can help a lot. Because once we involve people, they will not be surprised if there are some success stories, we can also take to them for them to learn." (Health Provider, IDI 1).

## Discussion

Reaching neglected populations remains a key barrier to the implementation of NTD control and elimination populations. Mobile and migrant populations continue to face unique challenges inhibiting access to health services including hydrocelectomies to patients in lymphatic filariasis endemic areas. As men their experiences with hydrocele and how they access health care are not described extensively in the literature [12]. Further, available equity analyses have largely focused on the implementation of preventive chemotherapy with lesser focus on the other strategies of disease control such as MMDP [13,14,39]. Our study sought to understand community health systems barriers that can affect the access of a case management intervention-hydrocele surgery for mobile and migrant hydrocele patients in Zambia. The key programmatic determinants identified include inadequate funding for hydrocele surgeries, absence of cross border collaborative structures, low levels of acceptability towards the surgery, poor fit between services and mobile and migrant populations and affordability of services. Whereas the key relational determinants included engagement of community leaders and community health workers as intermediaries, limited involvement of patient and patient groups and limited opportunities for collective action among the different community actors.

Sustainable control and elimination of neglected tropical diseases such as lymphatic filariasis is dependent on strong collaborations among actors working in endemic regions [40]. Collaborative involvement in implementation efforts can transform the relationships amongst the actors, improve the reach of services for populations with limited access to formal health systems, create new relationships/meanings for the intervention components within communities [41]. Dean et al write that insufficient funding and overreliance on donor funding for the delivery of NTD services weakens community based approaches [14]. Conceptualizations of CHS that have focus on CHWs programmes have limited the extent to which other community actors can be involved. Community based groups such as fishermen associations are not effectively engaged in the implementation of the services despite their potential to leverage their existing relationships with patients who are fishermen. Our findings echo those of Masong et al on the need to strengthen social networks which can enhance the implementation of surgeries in the district [13]. Strengthening actor relationships can be an opportunity to generate resources for the implementation of hydrocelectomies and other health services. According to Van Ryneveld, such new relationships have been effective in raising resources such as physical and financial assets, information and skills and linkages to other actors [42], useful in address health priorities identified by community members [43]. Further, increasing opportunities for communities monitoring or community involvement in decision making which was found to be minimal within our study site is critical for implementation success [44,45]. These strategies could increase the frequency with which community actors can meet and build their relationships and trust that is necessary for sustained engagement [43].

Active surveillance strategies that ensure that hydrocele patients are identified through reporting systems that collect adequate MMDP information still remain a barrier to access

[8,46]. Current case identification processes such as door to door approaches are not appropriate for mobile populations [39]. In our study, CHWs relied on their familiarity of the communities and their ability to cross over to neighboring countries to identify patients. Despite being absent in the district, cross border coordination mechanisms, are a feasible way of strengthening services [39]. Though the National NTD Masterplan calls for the provision of surgical services both in static health facility and through mobile outreach surgical camps [33], there use in the district has been minimal. Though the availability of surgical camps may not necessarily mean that hydrocele patients will utilize services, they can help to clear the backlog of surgeries [8] whilst addressing the challenges associated with getting transportation to the facility. As they can use already existing structures such as trained hydrocelectomy teams at local hospitals. Further the timing of the implementation of the surgical camps should also be done based on the migratory patterns of vulnerable populations. Patients ought to be regularly sensitized prior to the implementation of such camps through the different community health system structures. Allowing patients to adapt their plans to account for outreach activities which has been identified as a barrier in other settings [14].

Costs associated with accessing hydrocele surgeries within mobile and migrant populations has caused delayed or postponed treatments [47]. A potential intervention that could improve adoption of treatment is the provision of Conditional Cash Transfers which have been shown to have a positive impact on short term NTD outcomes such as treatment adherence in Leprosy particularly among more vulnerable populations [48]. Within the context of the implementation of hydrocele surgeries, the cash transfers can be used to provide support for patients where they can be cushioned from loss of income when recuperating or having adequate funding for transportation and other indirect costs associated with accessing services. Drawing on the different barriers and enablers that were identified during the assessment and recommendations provided by the study participants, Table 3 provides potential implementation strategies that can address them based on the ERIC taxonomy of implementation strategies and Proctor's guidelines on reporting on them [49,50].

## Limitations

The main limitation of the study is that at the time of data collection, there was very limited implementation of hydrocelectomies within the district due to insufficient funding. Consequently, the evaluation of community health system barriers is based the actors' previous implementation experiences in projects funded for short terms of between 1–2 years. By not being able to observe how the actors go about service delivery we could not a provide description of how these lenses were interlinked to the other CHS lenses.

## Conclusions

Community health systems provide a potential avenue through access to marginalised and vulnerable populations such as mobile and migrant populations can be improved. Our paper has identified potential CHS determinants such as inadequate funding for hydrocele surgeries, absence of cross border collaborative structures, poor fit of services, limited involvement of patient and patient groups and limited opportunities for collective action among the different community actors. Implementation strategies that can be employed to address these determinants include engagement of patient groups including fishing association, knowledge sharing across borders and creation of additional opportunities for community monitoring initiatives.

**Table 3. Potential implementation strategies to address identified CHS barriers.**

| CHS lens | Determinant | Implementation Strategy | Actor | Specific action | Action target | Temporality | Targeted outcomes |
|---|---|---|---|---|---|---|---|
| **Programmatic lens** | **Insufficient funding to effectively cover hydrocele patients in with limited involvement of local actors.** | **Access new funding sources** | **DHO officials National NTD Programme** | **Work with different community stakeholders to identify potential opportunities through which resources can be made available.** | **Community based organizations** | **Pre-implementation Initial implementation Full implementation Sustainment** | **Implementation Maintenance** |
| | Lack of continuity of care once the populations travel to neighboring countries. | Use of data warehousing techniques | DHO officials National NTD Programme Community leaders e.g. headmen & Chiefs. | Work with health officials across the borders to integrate clinical records to facilitate implementation across countries and enhance data collection | DHO officials National NTD Programme From neighboring countries | Pre-implementation Initial implementation Full implementation Sustainment | Implementation Maintenance |
| | Low levels of acceptability of surgery | Involvement of Patient groups | DHO officials Health providers CHWs NHCs | Engage fishermen associations and hydrocele patients who are members on the need to utilize the surgical services. | Fishermen associations Patient groups | Pre-implementation Initial implementation Full implementation Sustainment | Acceptability Adoption |
| | Limited sensitisation on available hydrocele services | Conduct educational meetings | DHO officials Health providers CHWs NHCs | Hold meetings and community dialogues targeted towards fishermen and migrants at specific points in the district such as harbors and fishing camps to teach about the value of the surgeries | Hydrocele patients | Full implementation Sustainment | Acceptability Adoption |
| | High costs associated with accessing services including loss of income/source of livelihood. | Alter allowance/incentive structures | Ministry of Health Implementing partners | Provide allowances such as transport allowance of livelihood support through programmes such as Conditional Cash transfer programmes that incentivize the adoption of hydrocele surgery | Hydrocele patients | Full implementation Sustainment | Acceptability Adoption |
| | Appointment systems lead to delays in accessing services | Adapt and Tailor strategies | DHO officials Health providers | Tailor the delivery of the surgeries such that migrants and mobile populations are prioritized when they go to health facilities to access services. | Hydrocele patients | Full implementation Sustainment | Adoption Implementation |
| **Relational lens** | Intermediaries such as CHWs, patients and Chiefs act as champions | Identify and prepare champions | DHO officials Health providers Previous hydrocele patients | Work with individuals dedicated to supporting implementation and resistance to the intervention within communities. | Opinion leaders within communities | Initial implementation Full implementation Sustainment | Adoption Implementation |
| | Limited involvement of community actors across the three countries | Capture and share local knowledge with community actors in the three countries to build stakeholder relationships | DHO officials National NTD Programme | Capture knowledge on how hydrocele services have been implemented in Luangwa and share it through dialogues with actors from other sites to facilitate implementation and sustainability. | DHO officials National NTD Programme From neighboring countries | Pre-implementation Initial implementation Full implementation Sustainment | Implementation Maintenance |

*(Continued)*

**Table 3.** (Continued)

| CHS lens | Determinant | Implementation Strategy | Actor | Specific action | Action target | Temporality | Targeted outcomes |
|---|---|---|---|---|---|---|---|
| Programmatic lens | **Insufficient funding to effectively cover hydrocele patients in with limited involvement of local actors.** | **Access new funding sources** | **DHO officials National NTD Programme** | **Work with different community stakeholders to identify potential opportunities through which resources can be made available.** | **Community based organizations** | **Pre-implementation Initial implementation Full implementation Sustainment** | **Implementation Maintenance** |
| | Few opportunities for community monitoring of services | Obtain and use patients and community members feedback on implementation | DHO officials Health providers Community actors Community members | Develop opportunities where patients/ Community members can be actively involved in their care, ask questions about the surgeries and provide feedback e.g. through community score cards | Hydrocele patients Community members | Pre-implementation Initial implementation Full implementation Sustainment | Implementation |

## Supporting information

**S1 File. Data collection tools.** Qualitative and quantitative data collection tools that were used in the study.
(DOCX)

**S2 File. GRAMMS reporting standard.** Filled out reporting guideline appropriate for a mixed methods study.
(DOCX)

**S3 File. Datasets.** Data collected and reported in the manuscript.
(ZIP)

## Acknowledgments

We would like to acknowledge Louis Adu-Amoah (ARNTD), the Luangwa District Health Office and Community Health Workers involved in the delivery of Lymphatic filariasis interventions in the district for facilitating the study processes.

## Author Contributions

**Conceptualization:** Patricia Maritim, Joseph Mumba Zulu.

**Data curation:** Patricia Maritim, Mwimba Chewe.

**Formal analysis:** Patricia Maritim, Mwimba Chewe, Margarate Nzala Munakaampe, Adam Silumbwe, George Sichone, Joseph Mumba Zulu.

**Funding acquisition:** Joseph Mumba Zulu.

**Investigation:** Patricia Maritim, Joseph Mumba Zulu.

**Methodology:** Patricia Maritim, Mwimba Chewe, Margarate Nzala Munakaampe, George Sichone, Joseph Mumba Zulu.

**Project administration:** Patricia Maritim, Adam Silumbwe, George Sichone, Joseph Mumba Zulu.

**Supervision:** Patricia Maritim, Joseph Mumba Zulu.

**Validation:** Joseph Mumba Zulu.

**Visualization:** Patricia Maritim, Mwimba Chewe, Margarate Nzala Munakaampe, George Sichone, Joseph Mumba Zulu.

**Writing – original draft:** Patricia Maritim, Mwimba Chewe, Margarate Nzala Munakaampe, Adam Silumbwe, Joseph Mumba Zulu.

**Writing – review & editing:** Patricia Maritim, Mwimba Chewe, Margarate Nzala Munakaampe, Adam Silumbwe, George Sichone, Joseph Mumba Zulu.

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
