## [Decision Letter · Decision Letter 0]

2 Nov 2022

PGPH-D-22-01419

Applying community health systems lenses to identify determinants of access to surgery among mobile & migrant populations with hydrocele in Zambia: a mixed methods assessment.

Dear Dr. Maritim,

Thank you for submitting your manuscript to PLOS Global Public Health. After careful consideration, we feel that it has merit but does not fully meet PLOS Global Public Health’s publication criteria as it currently stands. Therefore, we invite you to submit a revised version of the manuscript that addresses the points raised during the review process.

We look forward to receiving your revised manuscript.

Kind regards,

Abraham D. Flaxman, Ph.D.

Academic Editor

Journal Requirements:

1. Please include additional information regarding the survey or questionnaire used in the study and ensure that you have provided sufficient details that others could replicate the analyses. For instance, if you developed a questionnaire as part of this study and it is not under a copyright more restrictive than CC-BY, please include a copy, in both the original language and English, as Supporting Information.

Additional Editor Comments (if provided):

Reviewers' comments:

Reviewer's Responses to Questions

**Comments to the Author**

1. Does this manuscript meet PLOS Global Public Health’s publication criteria? Is the manuscript technically sound, and do the data support the conclusions? The manuscript must describe methodologically and ethically rigorous research with conclusions that are appropriately drawn based on the data presented.

Reviewer #1: Yes

2. Has the statistical analysis been performed appropriately and rigorously?

Reviewer #1: Yes

3. Have the authors made all data underlying the findings in their manuscript fully available (please refer to the Data Availability Statement at the start of the manuscript PDF file)?

Reviewer #1: Yes

4. Is the manuscript presented in an intelligible fashion and written in standard English?

Reviewer #1: Yes

5. Review Comments to the Author

Reviewer #1: Review of “Applying community health systems lenses to identify determinants of access to surgery among mobile & migrant populations with hydrocele in Zambia: a mixed methods assessment.”

The authors conducted a mixed-methods study to examine and understand determinants to adoption and integration of hydrocele surgeries among mobile/migrant populations in Zambia. It’s quite an impressive study with a large sample size. The quant and qual methods used seem to complement each other well. I think this study would be great for the journal with some fairly moderate revisions. Pending those, I would think a revised version would be fit for publication.

--

On a broad level, the paper is quite focused on mobile/migrant health but only around 20% of those sampled were migrants. The authors did say about 50% travel seasonally so this could perhaps just be stressed or better set up for that as the focused population in the discussion. I would argue that migrant populations (from other countries) and mobile populations (Zambians) might face different circumstances, though. As an aside, the discussion section alluded that some surveys were done in other countries. Is this true? If so, this should be better outlined in the methods section (see specific suggestions below on more information sampling requested).

Similarly, most people in this study were unemployed, yet the discussion focused on fisherman as a focused group. Could the authors focus their discussion more on the population studied?

A very minor percentage of patients underwent surgery. I wonder if the authors survey included more information on why that could complement the qualitative findings. I do wonder the relationship—or lack there of—of traditional healers and other stakeholders in the therapeutic landscape. Stacey Langwick writes a lot about this in Tanzania. Would recommend the authors to review some of her work.

CHWs have complex identities. Mohamed Yunus Rafiq writes about this in Tanzania. Would recommend the following study: https://pubmed.ncbi.nlm.nih.gov/31638989/

Rafiq also writes about use of religious leaders in public health campaigns. Would caution the authors to simply add that as a proposed solution without amore in-depth conversation for something that is complex on multiple levels: https://pubmed.ncbi.nlm.nih.gov/34915242/

A few other questions/comments

1) Was discrimination addressed on the survey?

2) The paper has multiple grammatical errors in the form of incomplete sentences.

3) Surgical camps are not a solution if people fear surgery or going for diagnosis. It’s a treatment that will not work if no one comes. Can the authors better lay out what this might look like as a solution or what it has looked like elsewhere?

4) Cash transfers a good idea.

5) Table 3 impossible to read. Recommend putting on landscape and reformatting.

Line by Line Comments

Abstract

- Recommend labeling sections of abstract (e.g. background, methods, results, conclusion)

- Line 25-27 the sentence “The absence of cross border collaborative structures that can support the continuity of care for patients moving across the three countries” is not a complete sentence. Recommend revising.

- Line 29, need a comma after “play”

Introduction

Line 52 – need comma after “higher level hospitals”

Lines 56-61. I don’t completely follow the authors explanation of how universal uptake of a treatment (such as hydrocelectomy) leads to health inequities, particularly point number three that delayed care leads to unequal treatment outcomes (this part makes sense) but then this has an impact on people’s social positions? How is that related to the intervention of surgery itself rather than this simply being a manifestation of the complex sociopolitical and economic circumstances within which people live?

The authors have provided an EXCELLENT discussion of the unique challenges faced by migrant/mobile populations. I would only ask them to also include language barriers as a major impediment to health, particularly in the context of surgery. This is very relevant to the topic at hand because sometimes distinguishing between an incarcerated hernia and a hydrocele can be difficult and good history taking is critical.

Line 95-97 – incomplete sentence.

More distinction on the programmatic and critical lens might be useful.

It might be helpful to provide more background specifically on hydrocele treatment in Zambia and provide the reader with background epidemiology and patterns of disease and treatment.

Methods

Line 112 – replace the word “with” with the word “and” so it is a complete sentence.

It would benefit the reader to understand, particularly for hydrocele treatment, at what level of the health system a hydrocelectomy would normally be offered? Is the lowest level that it is offered the district level hospital?

How many total households were visited to identify the 438 patients? How were households selected? Random sampling? Cluster sampling?

The authors should provide more detail on their sampling methods. How were original individuals chosen? How was sample size determined?

What online server was used?

Is the survey tool available to be published as supplementary material?

Were the qualitative interviews done in English? IF not, what language was used? Who and how were they translated?

It seems like the authors used primarily deductive coding with the proposed frameworks. Can the authors further specify why or how they used inductive coding?

Line 175 – not a complete sentence.

Results/Discussion

Line 185 – is this average age a mean or median? Can the authors provide standard error or deviation (if mean) or IQR if median and they want to provide the average. Otherwise would recommend just referring to the table.

Line 192 – Write 79% instead to be consteint with the other sentences. If authors want to avoid starting sentence with a number, consider revising the order of the sentence or simply combining with previous one.

Line 233-234 – I don’t follow the reason provided for patients not getting surgery if they knew it is available

Line 235 – why could migrants not access surgery? Several reasons were hypothesized in the introduction, but it is unclear if these reasons were elicited in the survey for migrants specifically.

Line 264 - is there a sample quote to back up this claim?

Line 284 – if patients from outside Zambia pay higher fee, would recommend authors provide more details on this in the intro since there is a big focus on migrants in their paper but not enough background on this in the introduction section.

The whole section seems to partly focus on implementation and how patients are not included but what interventions have really been proposed or used besides community health leaders trying to encourage people?

6. PLOS authors have the option to publish the peer review history of their article (what does this mean?). If published, this will include your full peer review and any attached files.

**Do you want your identity to be public for this peer review?** For information about this choice, including consent withdrawal, please see our Privacy Policy.

Reviewer #1: **Yes: **Zachary Obinna Enumah, MD PhD MA

---

## [Decision Letter · Decision Letter 1]

27 Mar 2023

PGPH-D-22-01419R1

Applying community health systems lenses to identify determinants of access to surgery among mobile & migrant populations with hydrocele in Zambia: a mixed methods assessment.

Dear Dr. Maritim,

Thank you for submitting your manuscript to PLOS Global Public Health. After careful consideration, we feel that it has merit but does not fully meet PLOS Global Public Health’s publication criteria as it currently stands. Therefore, we invite you to submit a revised version of the manuscript that addresses the points raised during the review process.

Overall, the reviewer feels that you have addressed their comments satisfactorily. However, the reviewer recommends that you address a few minor issues detailed in their comments.

Please note that we have only been able to secure a single reviewer to assess your manuscript. We are issuing a decision on your manuscript at this point to prevent further delays in the evaluation of your manuscript. Please be aware that the editor who handles your revised manuscript might find it necessary to invite additional reviewers to assess this work once the revised manuscript is submitted. However, we will aim to proceed on the basis of this single review if possible.

We look forward to receiving your revised manuscript.

Kind regards,

Alex Schaefer, PhD

Associate Editor

Journal Requirements:

2. Please include additional information regarding the survey or questionnaire used in the study and ensure that you have provided sufficient details that others could replicate the analyses. For instance, if you developed a questionnaire as part of this study and it is not under a copyright more restrictive than CC-BY, please include a copy, in both the original language and English, as Supporting Information."

Additional Editor Comments (if provided):

Reviewers' comments:

Reviewer's Responses to Questions

**Comments to the Author**

1. If the authors have adequately addressed your comments raised in a previous round of review and you feel that this manuscript is now acceptable for publication, you may indicate that here to bypass the “Comments to the Author” section, enter your conflict of interest statement in the “Confidential to Editor” section, and submit your "Accept" recommendation.

Reviewer #1: All comments have been addressed

2. Does this manuscript meet PLOS Global Public Health’s publication criteria? Is the manuscript technically sound, and do the data support the conclusions? The manuscript must describe methodologically and ethically rigorous research with conclusions that are appropriately drawn based on the data presented.

Reviewer #1: Yes

3. Has the statistical analysis been performed appropriately and rigorously?

Reviewer #1: Yes

4. Have the authors made all data underlying the findings in their manuscript fully available (please refer to the Data Availability Statement at the start of the manuscript PDF file)?

Reviewer #1: Yes

5. Is the manuscript presented in an intelligible fashion and written in standard English?

Reviewer #1: Yes

6. Review Comments to the Author

Reviewer #1: Recommend paper move forward with acceptance. Commend the authors for their time spent on the review. I would still ask that the additional information responded to in the comments on methods and some communities spanning two countries find its way into the final manuscript. I would also still ask the authors to comment one or two sentences to acknowledge that even if you bring surgical camps closer to a community---this does not necessarily translate to more people utilizing health services. Appreciate the opportunity to review an important contribution to this field and commend the authors on a well thought out study and well written manuscript.

7. PLOS authors have the option to publish the peer review history of their article (what does this mean?). If published, this will include your full peer review and any attached files.

**Do you want your identity to be public for this peer review?** For information about this choice, including consent withdrawal, please see our Privacy Policy.

Reviewer #1: **Yes: **Zachary Obinna Enumah

---

## [Editor Report · Decision Letter 2]

16 Jun 2023

Applying community health systems lenses to identify determinants of access to surgery among mobile & migrant populations with hydrocele in Zambia: a mixed methods assessment.

PGPH-D-22-01419R2

Dear Ms Maritim,

We are pleased to inform you that your manuscript 'Applying community health systems lenses to identify determinants of access to surgery among mobile & migrant populations with hydrocele in Zambia: a mixed methods assessment.' has been provisionally accepted for publication in PLOS Global Public Health.

Best regards,

Julia Robinson

Executive Editor